# PAR-AdvGAN: Improving Adversarial Attack Capability with Progressive Auto-Regression AdvGAN

## Abstract

Deep neural networks have demonstrated remarkable performance across various domains. However, they are vulnerable to adversarial examples, which can lead to erroneous predictions. Generative Adversarial Networks (GANs) can leverage the generators and discriminators model to quickly produce high-quality adversarial examples. Since both modules train in a competitive and simultaneous manner, GAN-based algorithms like AdvGAN can generate adversarial examples with better transferability compared to traditional methods. However, the generation of perturbations is usually limited to a single iteration, preventing these examples from fully exploiting the potential of the methods. To tackle this issue, we introduce a novel approach named Progressive Auto-Regression AdvGAN (PAR-AdvGAN). It incorporates an auto-regressive iteration mechanism within a progressive generation network to craft adversarial examples with enhanced attack capability. We thoroughly evaluate our PAR-AdvGAN method with a large-scale experiment, demonstrating its superior performance over various state-of-the-art black-box adversarial attacks, as well as the original AdvGAN. Moreover, PAR-AdvGAN significantly accelerates the adversarial example generation, i.e., achieving the speeds of up to 335.5 frames per second on Inception-v3 model, outperforming the gradient-based transferable attack algorithms. Our code is available at: `https://anonymous.4open.science/r/PAR-01BF/`

## 1 Introduction

Deep neural networks (DNNs) are widely used in different real-world applications, i.e., image classification (Li et al., 2020), emotional analysis (Qiang et al., 2020), and item recommendations (Pan et al., 2020). DNNs demonstrate human-surpassed performance when properly trained. However, DNNs can be vulnerable to adversarial examples crafted by attackers (Szegedy et al., 2013; Ma et al., 2021; Deng et al., 2020), which is a concern in safety-critical scenarios. Thus, a practical approach is to develop effective attack algorithms that can assess the robustness of DNNs against adversarial attacks at an early stage, ultimately enhancing model safety.

Currently, both white-box and black-box attack algorithms, such as gradient-based methods like FGSM (Goodfellow et al., 2014b), NAA (Zhang et al., 2022), SSA (Long et al., 2022), and optimization-based approaches such as PGD (Madry et al., 2017) and C&W (Carlini & Wagner, 2017), require continuous computation of the model's gradient information throughout the attack process. However, they all require extensive running time. Generative Adversarial Networks (GANs) (Goodfellow et al., 2014a) have demonstrated promising results for realistic sample generation by leveraging both generator and discriminator for training (Karras et al., 2021; Chang et al., 2022; Haidar & Rezagholizadeh, 2019; Croce et al., 2020). While the generator constructs high-quality examples, a discriminator learn to distinguish the original and generated examples. Furthermore, once the generator is trained, there will be no additional gradient computation for input examples.

As an early GAN-based model, AdvGAN incorporates a perturbation, denoted as $G(x)$, into the original image instance $x$ for attack (Xiao et al., 2018). AdvGAN aims to obtain the manipulated image $x + G(x)$ from the original instance $x$ through the discriminator. To achieve high attack success rates in both white-box and black-box attacks, AdvGAN introduces an adversarial loss on top

of GANs loss, ensuring the adversarial image is generated in a direction more effective for adversarial attacks. Additionally, it employs hinge loss to limit perturbation range, thereby preventing significant deviations between the adversarial and original images. Subsequently, AdvGAN++ (Jandial et al., 2019) further enhances the attack success rate by utilizing latent features instead of input image instances $x$. It optimizes the latent features during adversarial examples generation.

However, GAN-based methods suffer from several challenges. Both AdvGAN and AdvGAN++ generate perturbations in a single iteration, which limits their control over these perturbations. We observe the reason may be the generator continuously increases the perturbations during the iterative process (as illustrated in the **Appendix A.1**). This may not be effective against adversarial defenses and impacts the attack capability. It is critical since the goal is to maximize attack effectiveness with minimal perturbation. Furthermore, the transferability performance of such attacks is concerning, especially since internal model information is typically unavailable in real-world scenarios.

Inspired by recent works that utilise auto-regressive properties to generate realistic images or text (Chang et al., 2022; Yoon et al., 2019; Ni et al., 2020), we propose a novel GAN-based algorithm, Progressive Auto-Regression AdvGAN (PAR-AdvGAN) to generate adversarial examples with enhanced transferability. PAR-AdvGAN employs a progressive, auto-regressive iterative method to effectively capture the specific structures of input examples. This process gradually generates more diverse and realistic adversarial examples. Specifically, at time step $t$, we combine the input examples $x_{adv}^{t-1}$ from time step $t-1$ with the initial examples $x_0$ to generate the perturbation $G(x_{adv}^{t-1}, x_0)$ at time step $t$. Consequently, the manipulated examples shifts from $x + G(x)$ in the original AdvGAN to $x_{adv}^{t-1} + G(x_{adv}^{t-1}, x_0)$ in PAR-AdvGAN.

To achieve optimal performance with minimal perturbation, we propose $L_p$ loss to limit the perturbation range during iteration, thereby ensuring that the adversarial examples remain imperceptible to human. Furthermore, to enhance the quality of adversarial examples and mitigate potential distortions and significant noise during the generation process, we introduce $L_d$ loss, imposing a stringent constraint between the final adversarial examples $x_t$ and initial examples $x_0$. Finally, by independently optimising the generator and discriminator during training, we fine-tune the parameters of PAR-AdvGAN for more effective adversarial examples. Notably, owing to the high stealth and robust generalization capabilities, non-targeted adversarial attacks subject models to more rigorous evaluations and reveal more subtle vulnerabilities. Thus, we primarily focus on non-targeted adversarial attacks. We summarise the contributions as follows:

- We empirically study the limited transferability of adversarial examples generated by existing GAN-based algorithms. To address this, we explore the use of progressive generator network to enhance transferability.
- We propose an auto-regression iterative method and provide theoretical analysis on formulating $L_p$ and $L_d$ loss to ensure minimal distortions in the adversarial samples.
- Extensive experiments demonstrate that our PAR-AdvGAN significantly outperforms other methods, achieving highest attack success rates. Moreover, it outperforms traditional gradient-based transferable attack algorithms in both transferability and attack speed. We release the code of PAR-AdvGAN for future research development.

## 2 RELATED WORK

### 2.1 ADVERSARIAL ATTACKS

While numerous adversarial algorithms are dedicated to generating high-quality and robust adversarial samples, gradient-based attack algorithms constitute a main type. FGSM (Goodfellow et al., 2014b) was the first to utilise the model's gradients, which adds a small perturbation to the input data in the direction of the gradient, thereby maximising the loss function through gradient ascent to achieve optimal attack performance. MI-FGSM (Dong et al., 2018) incorporates a momentum factor in each iteration to mitigate the impact of local optima on the attack success rate. TI-FGSM (Dong et al., 2019) employs shifted images to calculate the input gradient, a process that involves convolving the original image's input gradient with a kernel matrix.

Other adversarial attack algorithms, such as PGD (Madry et al., 2017), project samples onto suitable attack directions and limit the size of perturbations to generate robust adversarial examples. C&W

method minimises the attack's objective function to optimise the generation process (Carlini & Wagner, 2017). AdvGAN (Xiao et al., 2018) employs an adversarial training process between the generator and discriminator. This process bolsters the generator's ability to produce adversarial samples, making them challenging for the discriminator to distinguish from genuine data. Besides attack purpose, we also note some other iterative training methods for GANs, such as the Progressive GAN (Karras et al., 2017) which divides the Generator into several layers, with each layer undergoing individual training. In our approach, we consider an auto-regression methods, where each subsequent generation is based on the results of previous step. Although auto-regression GAN has been improved for continuous generation tasks, all we need is the attack result of the last state, so we need to redesign it for this situation.

## 2.2 Adversarial Defenses

Adversarial defense represents an effective approach to mitigate the impact of attacks on DNNs. Commonly used adversarial defense techniques include denoising and adversarial training. The denoising technique employs preprocessing mechanisms to filter out adversarial examples, thereby preventing the poisoning of training data and reducing the likelihood of subsequent attacks on the model. Other notable works include HRGD (Liao et al., 2018), R&P (Xie et al., 2017) and so on (Dziugaite et al., 2016; Cohen et al., 2019).

Adversarial training enhances model robustness by incorporating adversarial examples into the training process. Ensemble adversarial training (Hang et al., 2020) works by decoupling the target model from adversarial examples generated by other black-box models, thereby defending against transferable attacks. To enhance the robustness of our algorithm against adversarial defenses, we validated the attack effectiveness of PAR-AdvGAN on the target model subjected to ensemble adversarial training.

## 3 Methodology

In this section, we first provide the problem definition of adversarial attacks. Then, we discuss the issue of perturbation escalation in AdvGAN and propose three strategies to optimise the generator, aiming to generate highly transferable adversarial samples. Finally, we provide a detailed implementation for the proposed PAR-AdvGAN method.

### 3.1 Problem Definition of Adversarial Attacks

Consider a clean data distribution $p_{data}$ in which benign samples are represented by $X \subseteq p_{data}$. In an untargeted attack, the network $f$ is misled by the manipulated sample $x_{adv}$. For the original sample $x \in X$, with the original label denoted as $m$, the adversarial goal can be defined as:

$$f(x_{adv}) \neq m \tag{1}$$

$$\|x_{adv} - x\|_n \leq \epsilon \tag{2}$$

where $\|\cdot\|_n$ represents the $n$-order norm (e.g., $L_2$ norm), and $\epsilon$ denotes the maximum perturbation.

### 3.2 Perturbation escalation in AdvGAN

AdvGAN adopts a non-repetitive iterative approach to improve attack performance. However, as iterations progress, the perturbation magnitude of the adversarial example increases rapidly. This issue arises because it treats each iterated example as an independent instance, neglecting its relation to the initial sample $x_0$. Additionally, AdvGAN fails to impose constraints on the distance between the generated samples and the initial samples. This suggests that the perturbations generated in each iteration will have significant magnitudes. To address this issue, we introduce three propositions:

**Proposition 1** The generator should obtain information about the original sample $x_0$.

**Proposition 2** To train the generative model, the inputs to the generator should include a significant number of non-initial samples, particularly those encountered during the adversarial process.

**Proposition 3** The generator should enforce constraints on the distance between adversarial samples and the initial sample $x_0$ throughout the iterative progress.

---

**Algorithm 1** Progressive Auto-Regression AdvGAN (PAR-AdvGAN)

---

**Input:** iteration number $T$, batch size $n$, progressive generator $G$, discriminator $D$, target network
$\quad$ $N$, corresponding label $m$, learning rate $\eta_1, \eta_2$, weight hyper-parameter $\lambda_1, \lambda_2, \lambda_3$
**Output:** $\theta_D, \theta_G$

1: **for** $i$ in (range) epoch **do**
2: $\quad$ Sample a mini-batch of $n$ examples
$\quad\quad$ $x = \{x(1), ..., x(n)\};$
3: $\quad$ $x_0 = x$
4: $\quad$ **for** $t = 1, ..., T$ **do**
5: $\quad\quad$ $P_t = G(x_{adv}^{t-1}, x_0)$, here $x_{adv}^0 = x_0$
6: $\quad\quad$ $x_{adv}^t = clip(x_{adv}^{t-1} + P_t)$
7: $\quad\quad$ **for** $k = 1, ..., S_d$ **do**
8: $\quad\quad\quad$ $g_{\theta_D} = \nabla_{\theta_D} L_D$
9: $\quad\quad\quad$ Update the discriminator by descending its stochastic gradient $S_d$ times:
10: $\quad\quad\quad$ $\theta_D = \theta_D - \eta_1 \cdot g_{\theta_D}$
11: $\quad\quad$ **end for**
12: $\quad\quad$ **for** $k = 1, ..., S_g$ **do**
13: $\quad\quad\quad$ $L_{adv} = -Cross\,Entropy(x_{adv}^t, m)$
14: $\quad\quad\quad$ $L_p = \|P_t\|_2$
15: $\quad\quad\quad$ $L_d = \|x_{adv}^t - x_0\|_2$
16: $\quad\quad\quad$ $L_G = (1 - D(x_{adv}^t))^2$
17: $\quad\quad\quad$ $g_{\theta_G} = \nabla_{\theta_G}(L_G + \lambda_1 L_p + \lambda_2 L_d + \lambda_3 L_{adv})$
18: $\quad\quad\quad$ Update the generator by descending its stochastic gradient $S_g$ times:
19: $\quad\quad\quad$ $\theta_G = \theta_G - \eta_2 \cdot g_{\theta_G}$
20: $\quad\quad$ **end for**
21: $\quad$ **end for**
22: **end for**
23: **return** $\theta_D, \theta_G$

---

### 3.3 PROGRESSIVE AUTO-REGRESSION ADVGAN

In this section, we first introduce the solutions for three propositions, namely progressive generator network, auto-regression iterative method, and generator constraints. Next, we explain the training processes for both the discriminator and the generator. Finally, as shown in Algorithm. 1, we provided the pseudo-code for the PAR-AdvGAN approach.

#### 3.3.1 PROGRESSIVE GENERATOR NETWORK

For Proposition 1, we adjust the generator to include initial example $x_0$ as an input, resulting in a revised generator $G(x_{adv}^t, x_0)$. To do this, we expand the channel dimension of the generator's first layer, and employ a concat operator to merge $x_{adv}^t$ and $x_0$ along the channel dimension. This design enables the generator to leverage information from both the current adversarial example $x_{adv}^t$ and initial input $x_0$, thus facilitating the generation of incremental adversarial perturbations during the iterative process (refer to line 5 in Alg. 1).

#### 3.3.2 AUTO-REGRESSION ITERATIVE METHOD

In Proposition 2, during each training iteration, we utilise a hyperparameter $T$ to regulate the number of interactions for $x_{adv}^t$ instances (refer to line 4). We iteratively generate $x_{adv}^t$ by adding perturbations $G(x_{adv}^{t-1}, x_0)$ to the preceding $x_{adv}^{t-1}$ (see line 6), then use the resulting gradient progression to update and train the generator.

$$\nabla_\theta = \frac{\partial L_{adv}}{\partial x + G(x)} \cdot \underbrace{\frac{\partial x + G(x)}{\partial G(x)}}_{1} \cdot \frac{\partial G(x)}{\partial \theta} \tag{3}$$

To better understand the auto-regression iterative progress, we decompose $\nabla_\theta$ in Eq. 3. Here, $\frac{\partial x + G(x)}{\partial G(x)}$ equals 1, so it can be omitted (See **Appendix A.3** for detailed proof). Following this, we further explore the relationship between $\frac{\partial G(x)}{\partial \theta}$ and $\frac{\partial L_{adv}}{\partial x + G(x)}$. Thus, $\frac{\partial G(x)}{\partial \theta}$ represents the degree of change in $G(x)$ with respect to changing in $\theta$. $\frac{\partial L_{adv}}{\partial x + G(x)}$ can be interpreted as the degree of change in $L_{adv}$

when changing $x + G(x)$. Therefore, we can interpret the gradient ascent process of the parameter $\theta$ as a modification of $\theta$ to drive $x + G(x)$ able to obtain a better adversarial effect. At this point, to enable $G$ to iteratively generate perturbations, we will replace $x$ with $x_{adv}^t$. This transforms the first part of Eq. 3 into $\frac{\partial L_{adv}}{\partial x_{adv}^t + G(x_{adv}^t)}$, indicating that $G$ continues to generate perturbations based on $x_{adv}^t$.

Given a network $N$ that accurately maps image $x$ sampled from the distribution $p_{data}$ to its corresponding label $m$. The adversarial sample $x_{adv}^t$ at time $t$ can be expressed as:

$$P_t = G(x_{adv}^{t-1}, x_0) \tag{4}$$

$$x_{adv}^t = clip(x_{adv}^{t-1} + P_t) \tag{5}$$

such that

$$N(x_{adv}^t) \neq m \tag{6}$$

Here, $P_t$ is the perturbation at time $t$, thus we have $x_{adv}^1 = x_0 + P_1$. And $G(\cdot, x_0)$ is the progressive generator network.

**Training of the Discriminator**  We train the discriminator to accurately distinguish adversarial samples generated by the progressive generator and actual samples from the data distribution $p_{data}$. Specifically, we fix the parameters related to the progressive generator and trained the discriminator $S_d$ times (line 7). The loss function $L_D$ can be written as:

$$L_D = (1 - D(x))^2 + D(x_{adv}^t)^2 \tag{7}$$

It is worth noting that we did not choose to calculate $L_D$ in the form of $log(1 - D(x))$ as in AdvGAN. This is because we find that the gradient of $log(1 - D(x))$ for $D$ will be very large and not smooth when $D(x)$ is close to 1, and gradient explosion will occur during iteration. We employ a squared form in the Eq. 10 to help mitigate this issue.

Hence, the gradient of the discriminator with respect to the parameters $\theta_D$ can be expressed using Eq. 8 (line 8):

$$g_{\theta_D} = \nabla_{\theta_D} L_D \tag{8}$$

By updating $\theta_D$ through gradient descent, we finally obtain the optimal parameters for the discriminator (line 9-10):

$$\theta_D = \theta_D - \eta_1 \cdot g_{\theta_D} \tag{9}$$

Here $\eta_1$ is the learning rate in discriminator training.

**Constraints on the Generator**  We propose the use of $L_{adv}$ to measure whether the adversarial samples are generated in a direction more conducive to the attack (refer to line 13).

$$L_{adv} = -Cross\ Entropy(x_{adv}^t, m) \tag{10}$$

It is worth noting that, in untargeted attacks, a larger value of $Cross\ Entropy(x_{adv}^t, m)$ indicates a more effective adversarial example. Consequently, to enhance the adversarial nature of $x_{adv}^t$ during gradient descent on $L_{adv}$, we prepend a negative sign to $Cross\ Entropy(x_{adv}^t, m)$. It is also feasible to replace $Cross\ Entropy$ with the loss function used in C&W (Carlini & Wagner, 2017).

To prevent the issue of perturbation explosion in the auto-regression iterative process, we introduce $L_p$ to constrain the magnitude of perturbation (refer to line 14), where $\|\cdot\|_2$ stands for the $l_2$ norm:

$$L_p = \|P_t\|_2 = \left\|G(x_{adv}^{t-1}, x_0)\right\|_2 \tag{11}$$

To fulfill Proposition 3, we introduce an additional loss function $L_d$ that enforces the generated adversarial examples to remain close to the initial example. This constraint ensures that the iterative progress of generating adversarial perturbations does not deviate significantly from the original input, thus maintaining the adversarial samples' proximity to the initial data (see line 15).

$$L_d = \left\|x_{adv}^t - x_0\right\|_2 \tag{12}$$

**Training of the Progressive Generator**    We train the progressive generator to generate adversarial samples with high transferability and low distortions from original samples while attacking the target neural network $N$. Specifically, we fixed the parameters related to the discriminator and trained the progressive generator $S_g$ times (refer to line 12).

As shown in Eq. 14, we computed the gradient of the progressive generator with respect to the parameters $\theta_G$ (line 16):

$$L_G = (1 - D(x_{adv}^t))^2 \tag{13}$$

$$g_{\theta_G} = \nabla_{\theta_G}(L_G + \lambda_1 L_p + \lambda_2 L_d + \lambda_3 L_{adv}) \tag{14}$$

Note that $L_G$ is the loss function to deceive the discriminator and $\lambda_1$, $\lambda_2$, $\lambda_3$ are the weight hyper-parameters that control the balance between loss functions. By updating $\theta_G$ through gradient descent, we ultimately obtain the optimal parameters for the progressive generator (line 17-18):

$$\theta_G = \theta_G - \eta_2 \cdot g_{\theta_G} \tag{15}$$

Here $\eta_2$ is the learning rate in discriminator training.

## 4    EXPERIMENTS

In this section, we present the experiments conducted to evaluate the performance of our method. To guide the analysis, we address the following research questions.

- What is the attack success rate of PAR-AdvGAN compared to the baseline AdvGAN? (**RQ1**)

- How does PAR-AdvGAN's performance in attack transferability and attack speed compare to state-of-the-art methods in adversarial attacks? Is it effective? (**RQ2**)

- Why does PAR-AdvGAN work effectively? (**RQ3**)

### 4.1    EXPERIMENT SETUP

#### 4.1.1    DATASET AND MODELS

We conducted the experiments on the ImageNet-compatible dataset consisting of 1000 images with a resolution of 299×299×3 (Papernot et al., 2018) [1]. The dataset generation process follows the literature (Dong et al., 2018; 2019; Gao et al., 2021; 2020)

Here we refer to the typical and state-of-the-art transferable adversarial attack methods (Xiao et al., 2018; Xie et al., 2019; Dong et al., 2019; 2018; Lin et al., 2019; Zhang et al., 2022). To ensure experiment fairness, we selected representative models from two types: normally-trained and defense-trained models. The normally trained models include Inceptionv3 (Inc-v3) (Szegedy et al., 2016), Inception-v4 (Inc-v4) (Szegedy et al., 2017), Inception-ResNet-v2 (IncRes-v2) (Szegedy et al., 2017), ResNet-v2-50 (Res-50) (He et al., 2016a;b), ResNet-v2-101 (Res-101) (He et al., 2016a;b), and ResNet-v2-152 (Res-152) (He et al., 2016a;b). As for the defense-trained models through ensemble adversarial training, we selected Inc-v3ens3 (Tramèr et al., 2017), Inc-v3ens4 (Tramèr et al., 2017), and IncResv2ens (Tramèr et al., 2017).

#### 4.1.2    BASELINE METHODS

We employ the original AdvGAN (Xiao et al., 2018) algorithm as our baseline to validate the transferability performance by incorporating self-regressive iteration in PAR-AdvGAN. Meanwhile, to evaluate our proposed PAR-AdvGAN, we selected seven state-of-the-art black-box adversarial attack methods as our competitive baselines, including FGSM (Goodfellow et al., 2014b), BIM (Kurakin et al., 2018), PGD (Madry et al., 2017), DI-FGSM (Xie et al., 2019), TI-FGSM (Dong et al., 2019), MI-FGSM (Dong et al., 2018), and SINI-FGSM (Lin et al., 2019).

---

[1]`https://github.com/cleverhans-lab/cleverhans/tree/master/cleverhans_v3.1.0/examples/nips17_adversarial_competition/dataset`

### 4.1.3 PARAMETER SETTINGS

All experiments in this study are conducted using the Nvidia RTX 6000 Ada 48GB. In all experiments, we set the following fixed parameters for each algorithm according to the settings in Kim (2020). For AdvGAN and PAR-AdvGAN, the training epochs are set to 60. The initial learning rate for both the Generator and Discriminator is set to 0.001, which is then reduced to 0.0001 at the 50th epoch. For DI-FGSM, we set the decay to 0, the resize_rate to 0.9, and the diversity_prob to 0.5. For TI-FGSM, decay is set to 0, kernel_name is set to "gaussian," len_kernel is set to 15, resize_rate is set to 0.9, and the diversity_prob is set to 0.5. For MI-FGSM, decay is set to 1. For SINI-FGSM, decay is set to 1, and $m$ is set to 5.

### 4.1.4 METRICS

Attack success rate (ASR) is a metric to evaluate the transferability of attacks. It quantifies the average proportion of mislabeled samples among all generated samples after the attack. Thus, a higher attack success rate signifies better transferability. Additionally, we use Frames Per Second (FPS) to assess the attack speed. Another crucial measure, the perturbation rate, is utilised to ensure that the adversarial images do not largely diverge from the original images in visual perception. A low value of this rate suggests that the adversarial examples maintain close visual fidelity to their originals. Detailed formulas are in the **Appendix A.4**.

### 4.2 RQ1: ATTACKING PERFORMANCE

As shown in Table. 1, we compare the attack success rates of the original AdvGAN and our proposed PAR-AdvGAN at three different perturbation rates of 8, 9, and 10. The comparisons are conducted using Inc-v3 and Inc-v4 as surrogate models and attacks are lunched on IncRes-v2. The results indicate that in most cases, our algorithm outperforms AdvGAN in terms of attack success rate.

We can see that compared to the most representative AdvGAN algorithm, PAR-AdvGAN has made significant improvements for attacking performance at a low perturbation rate. Specifically, similar to AdvGAN, PAR-AdvGAN, as a generative model, does not require additional gradient calculations based on different input data after training the generator. Compared to traditional gradient-based black-box transferable attack methods, it possesses faster attack speed. Therefore, we consider the PAR-AdvGAN algorithm feasible and suitable for attack scenarios that demand high transferability and fast generation of adversarial samples.

Table 1: ASR (%) of AdvGAN and PAR-AdvGAN on IncRes-v2. The adversarial examples are crafted on Inc-v3 and Inc-v4.

| Model | Attack | IncRes-v2 |
|-------|--------|-----------|
| Inc-v3 | AdvGAN | 13.9/38.9/43.2 |
| | **PAR-AdvGAN** | **27.1/35.2/41.4** |
| Inc-v4 | AdvGAN | 6.1/9.6/15.9 |
| | **PAR-AdvGAN** | **21.4/30.8/34.7** |

### 4.3 EFFECTIVENESS EXPERIMENT FOR RQ2: TRANSFERABILITY AND ATTACK SPEED

To validate the transferability and attack speed of PAR-AdvGAN compared to other SOTA methods, we conduct the experiments using various attack methods on Inc-v3, Inc-v4, and IncRes-v2 as source models to generate adversarial samples. We then conduct transferable attacks on different target models and use ASR and FPS as the main metrics, to validate the effectiveness of our algorithm.

Table 2: The attack success rates (%) on four undefended models and three adversarial trained models by various transferable adversarial attacks. The adversarial examples are crafted on Inc-v3. The best results are in bold.

| Model | Attack | Perturbation | Inc-v4 | Res-50 | Res-101 | Res-152 | Inc-v3ens3 | Inc-v3ens4 | IncResv2ens | mASR |
|-------|--------|--------------|--------|--------|---------|---------|------------|------------|-------------|------|
| Inc-v3 | AdvGAN | 8.41/9.88/10.26 | 32.9/61.9/65.9 | 42.6/77.7/82.8 | 47.8/75.3/83.1 | 38.8/66.6/72.8 | 15.1/44.0/52.5 | 30.2/54.5/63.0 | 9.9/25.7/26.5 | 31.04/57.95/63.80 |
| | PAR-AdvGAN | **8.29/9.27/9.95** | 53.7/63.1/68.4 | **74.8/85.0/89.8** | 79.3/86.2/89.5 | 68.2/78.0/82.5 | **39.4/53.5/63.8** | **45.7/60.8/69.4** | **28.0/39.0/46.5** | **55.58/66.51/72.84** |
| | FGSM | 8.79/9.74/10.69 | 26.2/28.0/30.3 | 26.2/29.0/30.6 | 23.3/25.8/27.6 | 22.8/24.1/27.0 | 13.8/14.5/15.5 | 14.0/14.3/14.3 | 6.0/6.1/6.1 | 18.9/20.25/21.62 |
| | BIM | 8.46/9.50/9.96 | 47.6/52.2/56.6 | 42.1/45.8/48.5 | 36.7/40.6/42.9 | 35.6/39.2/39.3 | 14.4/16.0/15.8 | 14.1/14.6/14.5 | 8.5/8.1/8.4 | 28.42/30.92/32.28 |
| | PGD | 8.76/9.79/10.35 | 44.6/46.2/50.2 | 38.6/40.7/43.5 | 33.1/35.5/37.3 | 28.9/34.5/35.8 | 12.4/14.2/14.4 | 12.4/13.3/13.1 | 6.8/7.4/7.7 | 25.25/27.40/28.85 |
| | DI-FGSM | 8.51/9.56/10.02 | 66.0/70.0/70.9 | 54.0/60.0/61.4 | 50.7/55.1/55.2 | 49.3/53.7/52.7 | 19.1/19.6/20.0 | 18.4/20.4/19.1 | 10.5/11.0/11.0 | 38.28/41.40/41.47 |
| | TI-FGSM | 8.60/9.65/10.10 | 52.4/55.3/56.5 | 39.7/45.2/45.2 | 34.4/39.9/41.1 | 34.8/39.0/43.1 | 31.6/34.8/35.3 | 34.1/37.7/39.2 | 21.4/25.9/25.8 | 35.48/39.68/40.88 |
| | MI-FGSM | 8.98/9.77/10.55 | 44.5/47.9/51.4 | 39.9/41.7/45.1 | 36.5/38.8/41.0 | 34.3/37.8/38.9 | 16.3/17.1/16.8 | 15.6/14.9/16.2 | 6.5/7.7/7.4 | 27.65/29.41/30.97 |
| | SINI-FGSM | 8.99/9.77/10.55 | 56.0/59.7/64.2 | 53.8/58.0/62.5 | 47.2/51.1/55.2 | 45.6/50.7/53.8 | 24.6/24.4/26.4 | 23.9/23.8/25.9 | 11.0/12.0/12.6 | 37.44/39.95/42.94 |

Table 3: The attack success rates (%) on four undefended models and three adversarial trained models by various transferable adversarial attacks. The adversarial examples are crafted on Inc-v4. The best results are in bold.

| Model | Attack | Perturbation | Inc-v3 | Res-50 | Res-101 | Res-152 | Inc-v3ens3 | Inc-v3ens4 | IncResv2ens | mASR |
|---|---|---|---|---|---|---|---|---|---|---|
| | AdvGAN | 9.64/11.67/12.11 | 55.6/75.9/59.1 | 48.6/77.5/70.6 | 54.4/74.8/69.5 | 40.4/67.0/58.4 | 13.4/26.5/24.4 | 20.3/54.6/43.0 | 16.0/25.7/21.1 | 35.52/57.42/49.44 |
| | PAR-AdvGAN | **9.55/11.05/11.60** | **72.9/85.6/87.8** | **66.1/79.6/85.0** | **73.7/86.4/89.8** | **55.7/69.2/74.8** | 13.0/17.4/20.2 | **35.0/50.9/58.2** | 15.7/25.6/30.2 | **47.44/59.24/63.71** |
| | FGSM | 9.74/11.64/12.58 | 28.4/31.9/32.9 | 23.7/26.3/28.4 | 21.1/24.2/25.4 | 20.6/24.3/25.6 | 13.1/13.2/13.4 | 10.9/11.7/12.3 | 5.9/6.6/6.8 | 17.67/19.74/20.68 |
| Inc-v4 | BIM | 9.98/11.46/11.91 | 60.1/62.5/62.4 | 42.5/45.8/46.2 | 38.9/40.2/41.0 | 34.7/39.8/39.8 | 13.9/15.7/16.1 | 13.5/15.7/14.8 | 9.3/10.9/9.6 | 30.41/32.94/32.84 |
| | PGD | 9.78/11.35/11.88 | 49.8/55.6/58.8 | 35.7/38.8/40.7 | 28.2/36.4/36.2 | 28.6/32.7/34.1 | 12.4/14.9/14.5 | 12.7/13.8/14.1 | 7.8/7.8/8.3 | 25.02/28.57/29.52 |
| | DI-FGSM | 10.01/11.49/11.94 | 75.4/80.4/80.3 | 57.6/63.7/62.9 | 51.2/57.5/56.5 | 49.9/55.0/55.7 | 18.5/19.6/20.6 | 16.4/18.5/18.7 | 10.7/13.5/12.4 | 39.95/44.02/43.87 |
| | TI-FGSM | 9.61/11.08/12.00 | 61.5/67.5/68.4 | 41.6/50.0/52.4 | 37.0/44.0/48.0 | 38.6/44.7/49.3 | **33.6/38.5/39.7** | 34.8/39.3/40.3 | **24.1/28.5/32.0** | 38.74/44.64/47.15 |
| | MI-FGSM | 9.81/11.42/12.22 | 53.2/61.2/61.7 | 40.2/43.2/47.0 | 36.7/41.6/43.9 | 34.0/39.8/41.3 | 15.0/15.6/16.4 | 14.6/15.3/15.0 | 6.2/7.9/7.6 | 28.55/32.08/33.27 |
| | SINI-FGSM | 9.79/11.39/12.18 | 75.1/78.2/80.1 | 63.1/69.0/70.6 | 58.3/65.9/66.7 | 56.9/62.6/64.8 | 27.8/31.1/32.1 | 26.4/28.8/29.9 | 14.3/16.6/17.4 | 45.98/50.31/51.65 |

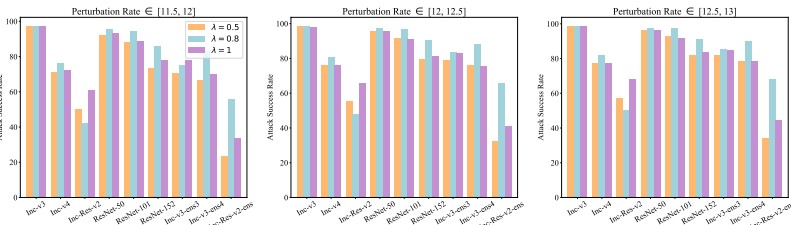

Figure 1: The performance of PAR-AdvGAN at different high perturbation rate intervals

### 4.3.1 EXPERIMENTS ON INC-V3

As shown in Table. 2, we conduct attacks using Inc-v3 as the source model with three different perturbation rates on target models of Inc-v4, Res-50, Res-101, Res-152, Inc-v3ens3, Inc-v3ens4, and IncRes-v2. We can see that our PAR-AdvGAN algorithm has achieved an average increase of 30.3% in attack success rate compared to other baselines. Moreover, despite DI-FGSM achieving better performance than PAR-AdvGAN on Inc-v4, which may be attributed to the randomness in model training, a comprehensive comparison across all models reveals that the attack success rate of PAR-AdvGAN is elevated by 24.6% compared to the best-performing competing baseline, DI-FGSM.

### 4.3.2 EXPERIMENTS ON INC-V4

As shown in Table. 3, we conduct attacks using Inc-v4 as the source model with three different perturbation rates on target models of Inc-v3, Res-50, Res-101, Res-152, Inc-v3ens3, Inc-v3ens4, and IncRes-v2. We can see that our PAR-AdvGAN algorithm has achieved an average increase of 20.13% in attack success rate compared to other baselines. Furthermore, compared to the best performing SINI-FGSM among competitive baselines, PAR-AdvGAN achieved an increase of 7.48% in ASR.

Table 4: The attack success rates (%) on four undefended models and three adversarial trained models by various transferable adversarial attacks. The adversarial examples are crafted on IncRes-v2. The best results are in bold.

| Model | Attack | Inc-v3 | Inc-v4 | Res-50 | Res-101 | Res-152 | Inc-v3ens3 | Inc-v3ens4 | IncResv2ens | mASR |
|---|---|---|---|---|---|---|---|---|---|---|
| | AdvGAN | 55.6/66.6/88.3 | 48.9/55.7/87.4 | 48.8/57.8/93.4 | 42.2/49.8/91.9 | 35.8/45.2/90.8 | 12.0/17.4/52.5 | 14.0/15.4/45.2 | 5.6/7.4/40.3 | 32.86/39.41/73.72 |
| | PAR-AdvGAN | 66.8/70.3/71.9 | **71.2/72.5/76.4** | **77.6/80.3/82.0** | 63.2/67.4/69.5 | **66.0/70.8/73.0** | 31.1/33.7/35.8 | 25.3/27.8/29.9 | 16.5/18.7/19.2 | 52.21/55.51/57.21 |
| | FGSM | 23.3/26.2/27.4 | 17.5/18.6/19.4 | 20.5/21.7/23.2 | 18.1/19.1/20.5 | 18.5/19.7/20.1 | 11.3/11.4/11.3 | 10.4/10.9/11.1 | 6.1/6.3/6.1 | 15.71/16.73/17.38 |
| IncRes-v2 | BIM | 58.4/62.0/63.8 | 47.1/51.8/40.7 | 41.9/46.4/47.3 | 36.6/39.4/42.2 | 35.6/38.7/41.8 | 14.9/14.7/15.2 | 12.8/13.8/14.6 | 9.9/10.4/11.2 | 32.15/34.70/35.85 |
| | PGD | 51.7/55.9/55.2 | 40.6/42.2/44.9 | 35.4/38.0/39.3 | 31.3/34.0/34.2 | 29.3/31.4/31.8 | 14.4/14.2/13.2 | 12.2/13.1/14.2 | 7.7/8.4/9.3 | 27.82/29.65/30.26 |
| | DI-FGSM | **79.5/79.0/79.3** | 69.7/72.0/74.8 | 61.8/61.7/67.2 | 58.3/59.6/62.5 | 57.2/56.9/59.9 | 20.9/20.5/22.2 | 18.9/20.3/20.9 | 14.8/15.1/14.7 | 47.63/48.13/50.18 |
| | TI-FGSM | 66.6/69.3/69.1 | 63.6/65.1/65.1 | 53.1/55.7/57.7 | 50.5/50.3/53.4 | 47.6/51.7/51.6 | **43.3/45.2/46.5** | **44.0/44.8/48.8** | **39.5/41.1/40.9** | 51.02/52.90/54.13 |
| | MI-FGSM | 58.2/59.8/61.7 | 49.8/50.1/55.7 | 43.7/46.1/48.4 | 39.2/44.7/43.3 | 37.6/39.5/42.6 | 15.0/16.1/17.4 | 14.8/15.4/15.3 | 8.9/9.4/9.8 | 33.40/35.13/36.77 |
| | **SINI-FGSM** | 76.7/79.3/80.6 | 70.1/73.8/75.7 | 66.1/70.4/73.0 | **63.8/66.7/71.0** | 60.9/63.9/66.6 | 34.7/35.2/37.0 | 29.5/29.9/31.4 | 20.6/20.7/22.4 | **52.80/54.98/57.21** |

### 4.3.3 EXPERIMENTS ON INCRES-V2

In this section, we conduct transferability tests on Inc-v3, Inc-v4, Res-50, Res-101, Res-152, Inc-v3ens3, Inc-v3ens4, and IncRes-v2 as target models with three different perturbation rates using IncRes-v2 as the source model. We have included the results in the Table 4. The results demonstrate that PAR-AdvGAN achieves an average increase of 14.96% in ASR compared to other baselines. We can see that although PAR-AdvGAN achieves a lower ASR of 0.02% than the best performing SINI-FGSM among competitive baselines, it outperforms AdvGAN by 6.31%.

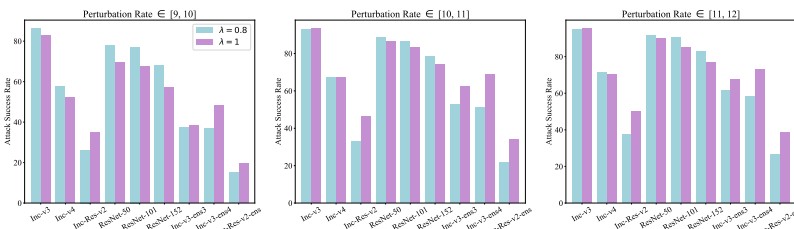

Figure 2: The performance of PAR-AdvGAN at different low perturbation rate intervals

Table 5: The attack success rates (%) on four undefended models and three adversarial trained models by various transferable adversarial attacks. The adversarial examples are crafted on ResNet-50. The best results are in bold.

| Model | Attack | Perturbation | Inc-v3 | Inc-v4 | Res-101 | Res-152 | Inc-v3ens3 | Inc-v3ens4 | IncResv2ens | mASR |
|---|---|---|---|---|---|---|---|---|---|---|
| | AdvGAN | 8.32/9.15/10.40 | 28.2/38.5/27.5 | 31.7/33.5/21.6 | 30.3/34.5/26.9 | 29.4/32.4/19.7 | 14/23.9/7.3 | 16.7/19.9/12.8 | 9.5/10.9/5.3 | 22.83/27.66/17.3 |
| | PAR-AdvGAN | **8.24/9.10/10.37** | **67.2/72.5/79.3** | 43.7/48.8/55.1 | **69.8/76.1/82.4** | 52.7/61.9/69.6 | **36.3/46.6/57.3** | **42.3/51.6/60.1** | **25.4/32/42.9** | **48.2/55.64/63.81** |
| | FGSM | 8.79/9.74/10.69 | 27/29.6/31.6 | 22.4/24.5/26 | 24.9/27.3/29.4 | 24.7/27/28.6 | 11.9/12.9/13.6 | 12.3/12/12.2 | 5.7/5.9/6.3 | 18.41/19.89/21.1 |
| ResNet-50 | BIM | 8.50/9.54/10.42 | 41.5/45.6/49.9 | 35.7/38.6/42.3 | 37/39.5/41.6 | 34/37.9/40.9 | 13.5/14/14.5 | 12/13.1/13.2 | 8.1/8/8.6 | 25.97/28.1/30.14 |
| | PGD | 8.34/9.37/10.46 | 30.8/35.6/39.2 | 26.4/30.3/33.1 | 26.9/29/32.5 | 24.5/27.9/31.5 | 11.6/11.6/13.1 | 10.4/11.8/12.6 | 6/6/7.5 | 19.51/21.74/24.21 |
| | DI-FGSM | 8.59/9.17/10.52 | 67.2/71.1/74.8 | **65.5/68.8/73.1** | 63.8/69.1/72.4 | **62.2/64.9/70.6** | 16.8/17/19 | 15/15/16.9 | 10.4/10.6/11.7 | 42.99/45.21/48.36 |
| | TI-FGSM | 8.45/9.46/10.38 | 51/54.9/60.1 | 48.9/54.9/58.2 | 44.9/47/52.2 | 42.7/45.8/51.1 | 34.4/36.5/38.2 | 35.4/39.1/41.2 | 24.7/28.1/31.3 | 40.29/43.76/47.47 |
| | MI-FGSM | 8.87/9.65/10.43 | 41.5/43.7/48.1 | 36.5/40/41.2 | 35.6/38.9/40.8 | 36.1/37.7/39.5 | 14.2/15/15.6 | 13.6/12.4/12.8 | 7.2/7.9/8 | 26.39/27.94/29.43 |
| | SINI-FGSM | 8.26/9.85/10.63 | 41.4/49.4/53.3 | 35.7/43.9/48.7 | 38.1/46.1/48.9 | 33.4/43.5/47.3 | 14.8/15.4/16.6 | 14.6/14/15.1 | 6.2/7.1/7.8 | 26.31/31.34/33.96 |

## 4.3.4 EXPERIMENTS ON RESNET-50

As shown in Table 5, we conduct attacks using ResNet-50 as the source model with three different perturbation rates on target models of Inc-v3, Inc-v4, Res-101, Res-152, Inc-v3ens3, Inc-v3ens4, and IncRes-v2. The experimental results demonstrate that PAR-AdvGAN consistently achieves superior performance across all target models compared to other baseline methods. Specifically, PAR-AdvGAN achieves the highest mASR of 48.2%, 55.64%, and 63.81% for the three perturbation rates, outperforming the best-performing baseline DI-FGSM by 5.21%, 10.43%, and 15.45%, respectively. Furthermore, PAR-AdvGAN shows significant improvements over AdvGAN, with an average increase in attack success rate of 25.37%. Although DI-FGSM achieves competitive performance on certain models such as Inc-v4 and Res-101, the overall effectiveness of PAR-AdvGAN across all models underscores its robustness and transferability.

## 4.3.5 EXPERIMENTS ON VIT-B/16

In Table 6, we present the results of attacks using ViT-B/16 as the source model with three different perturbation rates on several target models, including Inc-v3, Inc-v4, Res-50, Res-101, Res-152, Inc-v3ens3, Inc-v3ens4, and IncRes-v2. Unlike traditional convolutional neural networks (CNNs), Vision Transformers (ViTs) adopt a fundamentally different architecture for image classification tasks. Our experimental findings show that the proposed PAR-AdvGAN algorithm performs exceptionally well when transferred to ViT-based models, achieving an average increase of 6.85% in attack success rate (ASR) compared to AdvGAN across all target models. Specifically, PAR-AdvGAN consistently delivers the highest mean ASR values of 73.38%, 78.56%, and 81.63% across the three perturbation rates, surpassing all baseline methods, including DI-FGSM, which was the best performer in certain cases. These results underscore the robustness and transferability of PAR-AdvGAN, demonstrating

Table 6: The attack success rates (%) on four undefended models and three adversarial trained models by various transferable adversarial attacks. The adversarial examples are crafted on ViT-B/16. The best results are in bold.

| Model | Attack | Perturbation | Inc-v3 | Inc-v4 | Res-50 | Res-101 | Res-152 | Inc-v3ens3 | Inc-v3ens4 | IncResv2ens | mASR |
|---|---|---|---|---|---|---|---|---|---|---|---|
| | AdvGAN | 10.59/11.30/12.25 | 75.6/72/80.3 | 70.3/60.5/70.2 | 72.5/75.7/84.3 | 70.8/75.2/84.9 | 68.9/71.8/77.1 | 61.2/74.9/83.4 | 61.8/69.5/79.1 | 53.8/50/60.5 | 66.86/68.7/77.46 |
| | PAR-AdvGAN | **10.21/11.12/12.01** | **76.1/81.8/84.5** | 63.6/67/70.9 | **80.3/84.9/87.3** | **79.9/84.8/87.3** | **75.7/80.4/83** | **82.3/87.6/90.6** | **73.9/79.5/83.1** | **55.2/62.5/66.3** | **73.38/78.56/81.63** |
| | FGSM | 10.76/11.71/12.65 | 35/36.3/38.6 | 31.9/34.6/36.4 | 33.6/35.1/37.6 | 32.2/34/36 | 31.9/34/36 | 27.9/30.1/31.7 | 29.9/30.3/31.5 | 23.8/25.3/26.5 | 30.78/32.46/34.29 |
| ViT-B/16 | BIM | 10.90/11.47/12.37 | 55.5/58.7/60.7 | 50.5/49.7/54.1 | 54.2/55.6/59.2 | 48.5/51.4/56.8 | 46.6/47.6/54.4 | 39.2/38.5/42.4 | 39.7/41/42.8 | 30.5/32.4/35.1 | 45.59/46.86/50.69 |
| | PGD | 10.73/11.23/12.24 | 46.6/48/53 | 40.6/48.3/44 | 44.2/47/49.6 | 40.3/42.7/45.7 | 37.9/39.6/42.4 | 28.2/29.7/31.7 | 31.8/31.5/33.9 | 21.8/22.9/25.2 | 36.44/37.96/40.69 |
| | DI-FGSM | 10.59/11.58/12.04 | 75.6/77.8/78.9 | **70.3/74.1/75.1** | 72.5/77/74.9 | 70.8/74.5/73.1 | 68.9/71.8/71.9 | 61.2/67.7/67.4 | 61.8/65.8/67.5 | 53.8/58.5/57.9 | 66.86/70.9/70.84 |
| | TI-FGSM | 10.77/11.21/12.23 | 60.5/61/65.4 | 54.1/56.2/59.9 | 53.1/55.4/60.2 | 52.6/54.6/58.3 | 50.7/54.8/59.3 | 56.4/58.6/61.2 | 59.4/62.4/63.6 | 52.1/53/57.7 | 54.86/57/60.7 |
| | MI-FGSM | 10.75/11.58/12.36 | 53.7/55.6/59.1 | 47.4/49/52.8 | 50.9/54.3/57.7 | 47.7/50.5/53.2 | 45.5/48.9/50.9 | 38.6/40.5/43.7 | 40.5/42.6/43.9 | 30.7/33.7/36.7 | 44.38/46.89/49.75 |
| | SINI-FGSM | 10.83/11.66/12.45 | 58.2/61.8/65.5 | 52.7/56.3/60.9 | 55.7/61.2/64.7 | 51.1/56.1/59.6 | 49.8/54.1/57.9 | 44.5/47.5/49.4 | 44.1/46.7/50.4 | 37.9/39.7/43.9 | 49.25/52.93/56.54 |

its ability to maintain high effectiveness not only with traditional CNNs but also with more recent transformer-based models like ViT, thus proving its versatility and reliability across different model architectures.

### 4.3.6 ATTACK TRANSFERABILITY RESULT ANALYSIS

With the results from Tables 2- 6, it can be observed that in most cases, our adversarial attack algorithm shows significantly improved transferability compared to the original AdvGAN, especially at lower perturbation rates. Additionally, compared to other competitive baselines, PAR-AdvGAN exhibits the best transferability. Notably, to ensure the fairness of the experiments, our algorithm was consistently compared with other methods for a lowest perturbation rate. In instances where the perturbation rates were higher, some algorithms did not exhibit a proportional increase in attack transferability. However, the transferability is overall improved.

### 4.3.7 ATTACK SPEED ANALYSIS

As shown in Table. 7, we evaluated the computational efficiency of PAR-AdvGAN and seven competitive baselines using Inc-v3, Inc-v4, and IncRes-v2 as source models. We use FPS as the metric for measuring attack speed. It can be observed that across Inc-v3, Inc-v4, and IncRes-v2, PAR-AdvGAN exhibits speed improvements of 61, 88.3, and 158.5 times over the slowest-performing PGD algorithm among the competitive baselines. Furthermore, in comparison to the fastest-performing FGSM algorithm among the competitive baselines, PAR-AdvGAN achieves speed enhancements of 1.9, 2.5, and 4.4 times, respectively. We assert that PAR-AdvGAN demonstrates significantly higher attack speed in comparison to traditional gradient-based transferable methods, while simultaneously achieving state-of-the-art transferability performance.

### 4.4 ABLATION EXPERIMENT FOR RQ3

We investigate the effects of parameter $\lambda$ on the attack transferability as it is an important parameter to control the perturbation range. Fig. 1 shows the performance of PAR-AdvGAN with Inc-v3 as the source model, with a fixed $\epsilon$ of 20, and $\lambda$ set to 0.5, 0.8, and 1 for different target models. At $\lambda$ of 0.5, the specific perturbation rates are 11.75, 12.56, and 12.89. At $\lambda$ of 0.8, the specific perturbation rates are 11.55, 12.59, and 12.90. At $\lambda$ of 1, the specific perturbation rates are 11.66, 12.45, and 12.81. We can see that at higher perturbation rate intervals, setting $\lambda$ to 0.8 achieves best transferability performance.

| Method | Inc-v3 | Inc-v4 | IncRes-v2 |
|---|---|---|---|
| FGSM | 176.5 | 116.7 | 76.1 |
| BIM | 10.6 | 6.5 | 4.1 |
| PGD | 5.5 | 3.3 | 2.1 |
| DI-FGSM | 10.8 | 6.6 | 4.1 |
| TI-FGSM | 10.6 | 6.5 | 4.2 |
| MI-FGSM | 42.7 | 26 | 16.4 |
| SINI-FGSM | 8.6 | 5.3 | 3.3 |
| PAR-AdvGAN | **335.5** | **291.3** | **332.8** |

Table 7: FPS comparison of PAR-AdvGAN with seven competitive baselines

Fig. 2 compares the results with fixed $\epsilon$ of 16 and $\lambda$ set to 0.8 and 1. At $\lambda$ of 0.8, the specific perturbation rates are 9.40, 10.60, and 11.20. At $\lambda$ of 1, the corresponding perturbation rates are 8.95, 10.88, and 11.40. For lower perturbation rates, setting $\lambda$ to 1 achieves the best transferability.

## 5 CONCLUSION

In this paper, we present a novel Progressive Auto-Regression AdvGAN (PAR-AdvGAN) algorithm to boost adversarial attack capability through iterative perturbations. Specifically, to address the perturbation escalation issue in AdvGAN, we first adopt a progressive generator network to incorporate the initial sample $x_0$ in the perturbation generation process. An auto-regression iterative method is then proposed to include non-initial sample information in generator training. Furthermore, we constrain the distance between initial samples and subsequent samples. Our extensive experimental results exhibit the superior attack transferability of our method. Moreover, compared with the state-of-the-art gradient-based transferable attacks, our method achieves an accelerated attack efficiency.

## CODE OF ETHICS AND ETHICS STATEMENT

All authors of this paper have read and adhered to the ICLR Code of Ethics, as outlined at `https://iclr.cc/public/CodeOfEthics`. We ensure that our research follows the principles of integrity, fairness, and respect. The methodologies proposed in this work do not involve human subjects, and no personal data is used in our experiments. The datasets and models employed are open-source, and the outcomes of the experiments are aimed at advancing adversarial attack research without promoting any harmful applications. There are no conflicts of interest or external sponsorship affecting this work. We have ensured compliance with ethical standards and legal regulations throughout the research process.

## REPRODUCIBILITY

In order to ensure the reproducibility of our results, we have provided detailed descriptions of the experimental setup in the main text. Our proposed PAR-AdvGAN method is described in Section 3, and the algorithms used are thoroughly outlined, including pseudo-code (Algorithm 1). We have also included hyperparameter settings and model architectures in Section 4.1.3. Additional details about the datasets, metrics, and baseline methods are available in the supplementary materials. Moreover, we have made our code and trained models available in an anonymous repository at `https://anonymous.4open.science/r/PAR-01BF/`, ensuring easy access for future research and reproduction of our results.

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

# A APPENDIX

In this supplementary material, we first show the perturbation comparison diagram of the original AdvGAN and PAR-AdvGAN in **Introduction**. Secondly, we introduced the optimization objectives for AdvGAN and detailed proof of the omission in **Methodology**. Furthermore, we state the specific formulas of Attack Success Rate (ASR) and Frames Per Second (FPS) in **Metrics**. Moreover, we list the specific data table of the experiments performed on **IncRes-v2**. Finally, we detail the specific parameters of each algorithm in **Effectiveness Experiments** for code reproduction.

## A.1 SCHEMATIC DIAGRAM OF PERTURBATION COMPARISON

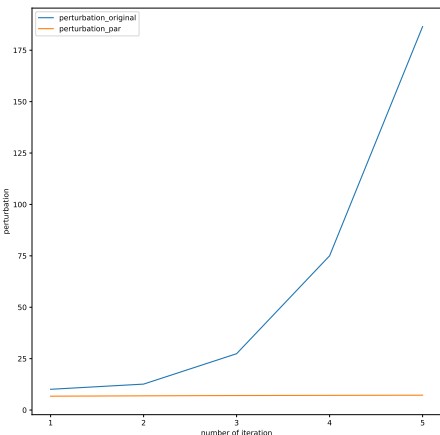

Figure 3: Schematic diagram of perturbation comparison (figure. 1 in the main paper)

## A.2 OPTIMIZATION OBJECTIVES FOR ADVGAN

As previously mentioned, in AdvGAN (Xiao et al., 2018), the adversarial sample $x_{adv}$ is synthesized from the original input $x$ and perturbation $G(x)$. By employing a process of iterative and competitive training between the generator $G$ and the discriminator $D$, AdvGAN can generate high-quality adversarial samples. This training approach enables AdvGAN to enhance the generator's capability to produce perturbations that effectively deceive the discriminator, leading it to classify these perturbations as real samples.

### A.2.1 OPTIMIZATION OBJECTIVE OF GENERATOR

For optimal attack performance, particularly in deceiving the target model $f$ to classify the sample as the target label $l$, AdvGAN utilises the loss function $L_{adv}$ to estimate the likelihood of successfully misleading $f$. In mathematical terms:

$$L_{adv} = E_x \left( J_f(x + G(x), l) \right) \tag{16}$$

Here, $E_x$ denotes the expected value of the input data $x$, in accordance with the distribution $p_{data}$. $J_f$ represents the loss function employed in training the target model $f$.

### A.2.2 OPTIMIZATION OBJECTIVE OF DISCRIMINATOR

The discriminator's role is to distinguish between adversarial samples and real samples. AdvGAN employs the loss function $L_{GAN}$ to maximize the distinction between manipulated data and the real data in discriminator $D$. The mathematical expression of $L_{GAN}$ is as follows:

$$L_{GAN} = E_x \left( log \left( 1 - D(x) \right) \right) + E_x \left( log D \left( x + G(x) \right) \right) \tag{17}$$

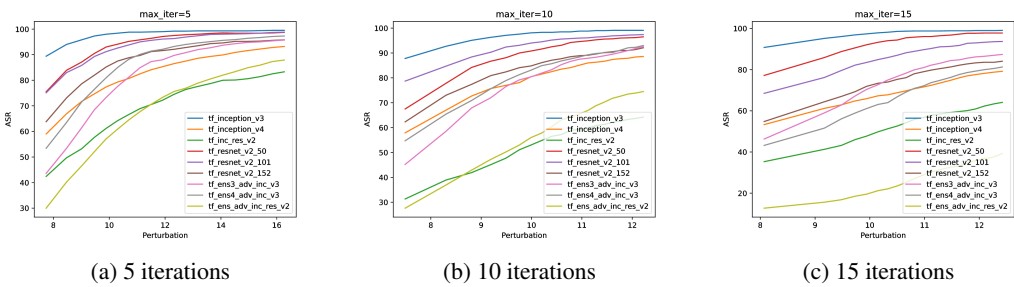

(a) 5 iterations      (b) 10 iterations      (c) 15 iterations

Figure 4: Ablation Experiment on the Number of Iterations

The term $E_x \left( log D \left( x + G(x) \right) \right)$ evaluates discriminator $D$'s capability to accurately identify adversarial samples. Conversely, the term $E_x \left( log \left( 1 - D(x) \right) \right)$ assesses the discriminator $D$'s inability to accurately identify original samples $x$.

### A.2.3 OPTIMIZATION OBJECTIVE OF PERTURBATIONS

Drawing on the findings from (Carlini & Wagner, 2017; Liu et al., 2016; Isola et al., 2017), AdvGAN incorporates the $L_{hinge}$ loss function to regulate the magnitude of the perturbations. With the incorporating of $L_{hinge}$, AdvGAN effectively limits the perturbation magnitude, ensuring that the generated adversarial samples remain imperceptible and closely resemble the original samples. $L_{hinge}$ is shown as:

$$L_{hinge} = E_x \left( max(0, \|G(x)\|_2 - c) \right) \tag{18}$$

$\|\cdot\|_n$ denotes the $L_2$ norm, and $c$ is a user-specified bound.

### A.3 DETAILED PROOF OF THE OMISSION IN EQ. 3

Through the chain rule of gradients, we can split $\bigtriangledown_\theta$ into $\frac{\partial L_{adv}}{\partial x+G(x)} \cdot \frac{\partial x+G(x)}{\partial G(x)} \cdot \frac{\partial G(x)}{\partial \theta}$. However, in order to explore the relationship between $\frac{\partial L_{adv}}{\partial x+G(x)}$ and $\frac{\partial G(x)}{\partial \theta}$, $\frac{\partial x+G(x)}{\partial G(x)}$ is redundant.

The reason is that, when calculating the partial derivative of $x + G(x)$ with respect to $G(x)$, $x$ is regarded as a constant and $G(x)$ is a variable. Therefore, changes in $x$ have no direct impact on changes in $G(x)$, and its derivative with respect to $G(x)$ is 0, while the derivative of $G(x)$ with respect to itself is 1. Hence, $\frac{\partial x+G(x)}{\partial G(x)}$ equals 1 and can be omitted. In this way, we have removed the redundant terms and unified the remaining two terms.

### A.4 FORMULAS OF ASR AND FPS

$$ASR = \frac{Number\ of\ misleading\ samples}{Number\ of\ adversarial\ samples} \tag{19}$$

$$FPS = \frac{Number\ of\ samples}{Running\ time\ of\ these\ samples} \tag{20}$$

# B ABLATION EXPERIMENT OF ITERATION NUMBER

## B.1 THE SPECIFIC PARAMETERS OF EACH ALGORITHM IN THE EFFECTIVENESS EXPERIMENT

### B.1.1 SPECIFIC PARAMETERS FOR EACH ALGORITHM ON INC-V3

The specific perturbation rates for PAR-AdvGAN were 8.2869, 9.2728, and 9.9546, while for AdvGAN they were 8.4143, 9.8761, and 10.2608. FGSM had specific perturbation rates of 8.7899, 9.7433, and 10.6936, BIM had rates of 8.4570, 9.4999, and 9.9611, and PGD had rates of 8.7574, 9.7935, and 10.3477. For DI-FGSM, the specific perturbation rates were 8.5112, 9.5550, and 10.0177, for TI-FGSM they were 8.6022, 9.6454, and 10.0996, for MI-FGSM they were 8.9799, 9.7688, and 10.5535, and for SINI-FGSM they were 8.9885, 9.7707, and 10.5512.

Furthermore, the specific parameter configurations for each algorithm under the setting of Inc-v3 are as follows: For AdvGAN, we set $\lambda$ to 0.1 for all three perturbation rates and eps to 10.0, 14.0, and 18.0 respectively. For PAR-AdvGAN, $\lambda$ is set to 1 for all three perturbation rates, and eps is set to 16.0. The number of iterative generations during training is set to 10, and the performance is evaluated separately for one, two, and three iterations of generation for each perturbation rate. For FGSM, we set eps to 9, 10, and 11. For BIM, eps is set to 17, 19, and 20. For PGD, eps is set to 16, 18, and 19. For DI-FGSM, eps is set to 17, 19, and 20. For TI-FGSM, eps is set to 17, 19, and 20. For MI-FGSM, we set eps to 11, 12, and 13. For SINI-FGSM, we set eps to 11, 12, and 13.

### B.1.2 SPECIFIC PARAMETERS FOR EACH ALGORITHM ON INC-V4

The specific perturbation rates for PAR-AdvGAN were 9.5509, 11.0456, and 11.6010, while for AdvGAN they were 9.6403, 11.6715, and 12.1057. FGSM had specific perturbation rates of 9.7433, 11.6408, and 12.5848, BIM had rates of 9.9811, 11.4568, and 11.9144, and PGD had rates of 9.7801, 11.3461, and 11.8804. For DI-FGSM, the specific perturbation rates were 10.0124, 11.4927, and 11.9433, for TI-FGSM they were 9.6131, 11.0770, and 11.9980, for MI-FGSM they were 9.8127, 11.4231, and 12.2215, and for SINI-FGSM they were 9.7882, 11.3879, and 12.1840.

Furthermore, the specific parameter configurations for each algorithm under the setting of Inc-v4 are as follows: For AdvGAN, we set $\lambda$ to 0.1 for all three perturbation rates and eps to 12.0, 18.0, and 14.0 respectively. For PAR-AdvGAN, $\lambda$ is set to 1 for all three perturbation rates, and eps is set to 16.0. The number of iterative generations during training is set to 10, and the performance is evaluated separately for one, two, and three iterations of generation for each perturbation rate. For FGSM, we set eps to 10, 12, and 13. For BIM, eps is set to 20, 23, and 24. For PGD, eps is set to 18, 21, and 22. For DI-FGSM, eps is set to 20, 23, and 24. For TI-FGSM, eps is set to 19, 22, and 24. For MI-FGSM, we set eps to 12, 14, and 15. For SINI-FGSM, we set eps to 12, 14, and 15.

### B.1.3 SPECIFIC PARAMETERS FOR EACH ALGORITHM ON INCRES-V2

The specific perturbation rates for PAR-AdvGAN were 9.8455, 10.5856, and 10.8302, while for AdvGAN they were 10.6939, 11.6411, and 12.5852. FGSM had specific perturbation rates of 10.6939, 11.6411, and 12.5852. BIM had rates of 10.0046, 11.0145, and 11.4804. PGD had rates of 10.3459, 10.8604, and 11.3555. For DI-FGSM, the specific perturbation rates were 10.0459, 11.0495, and 11.5136. For TI-FGSM, they were 10.1527, 10.5860, and 11.1589. For MI-FGSM, they were 10.5966, 11.4195, and 12.2130. For SINI-FGSM, they were 10.5448, 11.3682, and 12.1566.

Furthermore, the specific parameter configurations for each algorithm under the setting of IncRes-v2 are as follows: For AdvGAN, we set $\lambda$ to 0.1 for all three perturbation rates, and eps to 12.0, 18.0, and 28.0, respectively. For PAR-AdvGAN, $\lambda$ is set to 1 for all three perturbation rates, and eps is set to 20.0. The number of iterative generations during training is set to 10, and the performance is evaluated separately for one, two, and three iterations of generation for each perturbation rate. For FGSM, we set eps to 11, 12, and 13. For BIM, eps is set to 20, 22, and 23. For PGD, eps is set to 19, 20, and 21. For DI-FGSM, eps is set to 20, 22, and 23. For TI-FGSM, eps is set to 20, 21, and 22. For MI-FGSM, we set eps to 13, 14, and 15. For SINI-FGSM, we set eps to 13, 14, and 15.

## C    LIMITATIONS

We notice a slight difference in the attack success rate of PAR-AdvGAN on Inc-v3ens3 and In-cResv2ens compared to the best baseline TI-FGSM in Tab. 3, although for overall mASR our method achieves the best performance. This inconsistency suggests a possible dependency of our approach on model selection. The main reason for the inconsistency is related to whether the method requires further interaction with the model after training the Generator regarding the generation process. Our method PAR-AdvGAN doesn't require the interaction, while TI-FGSM needs to continuously interact with the model for the generation process to obtain gradient information. It is always be feasible for different models and we will consider reducing the dependence on model selection in the future to generate more transferable adversarial examples.

