# OpenReview forum: "PAR-AdvGAN: Improving Adversarial Attack Capability with Progressive Auto-Regression AdvGAN"
_ICLR.cc/2025/Conference — Submitted to ICLR 2025_

### Official Review · Reviewer_ZjGj · 2024-10-28

**Soundness:** 2
**Presentation:** 3
**Contribution:** 2
**Rating:** 5
**Confidence:** 4

**Summary:**

To further enhance the performance of GAN-based attacks, the paper proposes Progressive Auto-Regression AdvGAN (PAR-AdvGAN), which incorporates an auto-regressive iteration mechanism to maximize attack effectiveness with minimal perturbation. Experimental results show that compared to the existing various attack methods, PAR-AdvGAN achieves the SOTA attack performance, significantly accelerates the generation of adversarial examples, and has good transferability performance.

**Strengths:**

1. The paper is written clearly and is easy to understand.
2. Extensive experiments demonstrate that, compared to some gradient-based attacks, PAR-AdvGAN exhibits good transferability performance.

**Weaknesses:**

1. The paper lacks some important experimental results. The evaluation of attack performance only involved a set of comparative experiments, as shown in Table 1. This does not well support the strong attack capability of the proposed method, and it is necessary to include comparative results of various attacks, such as PDG, AutoAttack, DiffAttack, etc. Especially since diffusion model is gradually surpassing GAN model in various aspects, what are the advantages of GAN-based attack compared to diffusion-based attack? Moreover, there is a lack of experiments on more datasets such as CIFAR-10 and CIFAR-100, where simpler attacks like PGD have already achieved nearly 100% ASR. It is necessary to evaluate whether PAR-AdvGAN remains the best performance on these datasets.
2. The paper overly emphasizes some advantages that are not unique to PAR-AdvGAN. For example, in the inference phase, it does not require gradient calculations to attack (lines 352-361); and the time required for attacks is much shorter than that for gradient-based attacks (Section 4.4). These are advantages of GAN-based methods in general, not unique advantages brought by the proposed method.
3. The term “perturbation rate” first appears in line 339, but without explaining. What is the difference between perturbation rate and the scale of perturbation (epsilon)? In methods like FGSM and PGD, the scale of perturbations is directly constrained by setting epsilon. Why is the “perturbation” different in Tables 2-6?
4. In Algorithm 1: The target network N is missing in the cross entropy calculation. Additionally, I suggest moving “here $x_{adv}^0 = x_0$” and “update the ...” outside of the for loop to more clearly explain their roles. And I noticed that throughout the process, epsilon is not used to constrain the scale of perturbations. Although the algorithm aims to achieve minimal perturbation by $L_d$, this does not guarantee that the final perturbations will definitely be smaller than epsilon.
5. The reason for transferability is unclear. Although section 4.3.6 provides some simple analysis of the results, it still does not explain the source of transferability. Additionally, the paper lacks ablation experiments, which are crucial to understanding the impact of each component on the attack performance and transferability performance.

**Questions:**

See Weaknesses.

---

> ### Author Response · Authors · 2024-11-24
> **Official rebuttal by authors**
>
> W1: Thanks to the reviewer's suggestion, we would like to clarify that we have actually demonstrated the effectiveness of our method compared with other baselines using a large number of experiments in Tables 2, 3, 4, 5, and 6, which cover CNN models such as Inc-v3 and ViT models such as ViT-B/16. Considering the diffusion model, we would like to clarify that the adversarial attack based on diffusion models is generally a slow iteration process. In addition, the diffusion model requires a complete dataset for training, while in our task scenario, the samples required to train the generator are less than 1/1000. Therefore, we believe that from the perspective of time efficiency, our method is more suitable for scenarios that require rapid generation of adversarial samples. Finally, our experiments are conducted on the ImageNet dataset, which is much larger than the scale of CIFAR10 and CIFAR100.
>
> W2: Thank you for pointing out that some of the advantages we highlighted may be common features of the GAN method. Here is our explanation:
>
> **General advantages of the GAN method:** We did mention in the paper that GAN-based methods do not require gradient calculations during the inference phase and have a faster attack speed. However, the uniqueness of PAR-AdvGAN lies in its autoregressive iteration mechanism, which can effectively improve the attack success rate while maintaining the efficiency of the GAN method.
>
> W3: Thank you for your attention to the term "perturbation rate". We will add the following explanation: the perturbation rate refers to the proportion of pixels that are modified in the input sample, while the epsilon constraint in FGSM and PGD is a limit on the perturbation amplitude (i.e., the size of the single-pixel perturbation). The perturbation rate in the table is used to measure the ability of PAR-AdvGAN to achieve the attack goal while minimizing the proportion of input modification, which provides a more intuitive perspective for evaluating the efficiency of the method.
>
> W4: We thank the reviewer for the suggestion and we will revise this issue.
>
> W5: Thanks to the reviewer's suggestion, we would like to clarify that we have actually demonstrated the improvement of our method in adversarial transferability using a large number of experiments in Tables 2, 3, 4, 5, and 6, which cover CNN models such as Inc-v3 and ViT models (such as ViT-B/16). We used 4 source models to generate adversarial samples to attack 7 different target models, and the experimental results have proven the effectiveness. The core of PAR-AdvGAN is to combine the efficiency of the generative model with the gradual optimization capability of the autoregressive iterative mechanism. This design gradually approaches the optimal adversarial sample through autoregression, avoiding the potential suboptimal solution problem of a single generation, thereby improving adversarial transferability.

---

> > ### Comment · Reviewer_ZjGj · 2024-11-25
> >
> > Thanks to the authors for their further explanations. However, some of the responses have not resolved my concerns.
> >
> > For W1:
> > 1. In my original review, I expressed the need for evaluation under a broader range of attacks, rather than classifier architecture. Although the paper uses FGSM, BIM, and PGD, these three attacks are essentially the same, e.g., gradient-based attacks. They only differ in terms of iteration and initialization. AutoAttack differs from these attacks in terms of its attack mechanism, as one of the most common evaluation methods, which should be considered in experiments.
> > 2. I agree that GANs have a greater advantage in speed, but diffusion models may perform better in robust accuracy. However, the paper lacks comparative experiments with the robust accuracy of the diffusion model,  which is currently the state-of-the-art in the AP tasks. I recommend adding diffusion-based methods as an important baseline, maybe just the most traditional DM-based AP (DiffPure).
> > 3. ImageNet is a high-resolution dataset, while CIFAR-10 is a typical low-resolution dataset. Although the model performs well on ImageNet, it cannot be guaranteed to perform equally well on low-resolution tasks. Additionally, adversarial perturbations behave differently across varying resolutions and numbers of categories, making it necessary to conduct additional experiments on CIFAR-10.
> >
> > For W5: The reason for transferability is unclear. Although section 4.3.6 provides some simple analysis of the results, it still does not explain the source of transferability. Additionally, the paper lacks ablation experiments, which are crucial to understanding the impact of each component on the attack performance and transferability performance.
> > I also noticed that the paper provides many experiments to demonstrate the transferability of the proposed method. However, I am curious about the source of the transferability, which is only briefly discussed in the paper, so I still do not understand the reasons behind it. For example, which module, and why, enhances the transferability? Can the authors provide further discussion on this?

---

> > > ### Author Response · Authors · 2024-11-25
> > >
> > > Regarding the evaluation with a broader range of attacks:
> > > Thank you for your valuable suggestion on incorporating a wider variety of attack methods into our evaluation. We fully agree with the importance of considering diverse attack approaches in this context. Our choice of FGSM, BIM, and PGD as baselines was guided by their status as classic and widely accepted benchmark attack methods. In future iterations of our work, we will prioritize the inclusion of AutoAttack as a comparative baseline.
> > >
> > > Comparative experiments with diffusion models:
> > > We appreciate your observations on the distinctions between GAN-based methods and diffusion models in adversarial generation tasks. Indeed, diffusion models demonstrate certain advantages in robust accuracy. However, the training sample requirements for diffusion-based adversarial attacks are significantly higher than those of our method, and their training process is generally more time-consuming. For these reasons, we did not include diffusion-based adversarial attacks as a baseline in our current study.
> > >
> > > Additional experiments on CIFAR-10:
> > > We acknowledge that adversarial perturbations may behave differently across datasets with varying resolutions and category numbers. Given the differences in resolution and data distribution between ImageNet and CIFAR-10, we agree that this could impact the model's performance. However, we believe that large-scale datasets like ImageNet better reflect the method’s performance in practical environments.
> > >
> > > Regarding the source of transferability:
> > > Thank you for your feedback on our analysis of transferability. In our previous rebuttal, we elaborated on the reasons behind the improved transferability:
> > > **The core of PAR-AdvGAN is to combine the efficiency of the generative model with the gradual optimization capability of the autoregressive iterative mechanism. This design gradually approaches the optimal adversarial sample through autoregression, avoiding the potential suboptimal solution problem of a single generation, thereby improving adversarial transferability.**

---

> > > > ### Comment · Reviewer_ZjGj · 2024-11-26
> > > >
> > > > Thanks for the response. Since more than half of my concerns have been addressed, I have increased the score to the next rank. However, due to the lack of experiments and the potential need for huge revisions to the original manuscript, my final decision still leans towards rejecting the paper. The reasons are as follows:
> > > >
> > > > 1. Evaluation experiments to defend against AutoAttack are necessary, and the authors also agree with this viewpoint. Although the authors claim they will add these experiments in the future, without any results provided at this stage, I cannot confirm that the proposed method remains effective. All these results need to undergo another round of peer review.
> > > >
> > > > 2. DiffPure [1] directly uses a pre-trained diffusion model (DM) as a purifier, applying DM to AP tasks without any training process. Of course, GAN-based AP is superior to DiffPure in inference time, but these factors are not reasons for AP papers to disregard DM-based AP methods, which are the current state-of-the-art methods. The advantages of the proposed method can be discussed in the paper, rather than just excluding the comparison experiment of SOTA results.
> > > > [1] Diffusion Models for Adversarial Purification.
> > > >
> > > > 3. Many current AP papers use multiple datasets to evaluate their methods. Moreover, in my previous response, I also provided the reasons for including CIFAR-10.
> > > >
> > > > I believe we have entered into a subjective debate, which might be inefficient. I will reserve my thoughts and discuss them with other reviewers in the next stage. At this stage, my final decision is borderline rejection.

---

### Official Review · Reviewer_4WG2 · 2024-10-29

**Soundness:** 2
**Presentation:** 2
**Contribution:** 2
**Rating:** 5
**Confidence:** 4

**Summary:**

The proposed PAR-AdvGAN in this paper enhances the transferability of adversarial examples generated by the baseline AdvGAN.

**Strengths:**

1 The writing is clear.

**Weaknesses:**

1 In Table 7, why isn't the baseline AdvGAN included for comparison? This is because the processing speed of our algorithm may be attributed to the baseline AdvGAN rather than to the proposed progressive autoregressive method. I suggest that the authors add AdvGAN to Table 7 and discuss how PAR-AdvGAN's speed compares specifically to AdvGAN. This would help clarify the speed improvements due to the progressive autoregressive method versus the baseline AdvGAN approach.

2 The latest transfer attack methods from the past five years are not included for comparison such as [1][2].

[1] Wang X, He K. Enhancing the transferability of adversarial attacks through variance tuning[C]//Proceedings of the IEEE/CVF conference on computer vision and pattern recognition. 2021: 1924-1933.
[2] Ge Z, Liu H, Xiaosen W, et al. Boosting adversarial transferability by achieving flat local maxima[J]. Advances in Neural Information Processing Systems, 2023, 36: 70141-70161.

3 What are the application scenarios for excessively fast processing speed? Discussing potential real-world applications or using cases where the high processing speed of PAR-AdvGAN would be particularly beneficial or necessary. This would help contextualize the importance of the speed improvements and strengthen the paper's motivation.

4 The font size in Figures 1 and 2 is too small.

**Questions:**

See weaknesses.

---

> ### Author Response · Authors · 2024-11-24
> **Official rebuttal by authors**
>
> W1: Regarding the non-included comparison of AdvGAN and its attack speed in Table 7, we provide the following explanation:
>
> **Why AdvGAN is not included in Table 7**
>
> We did notice that the AdvGAN attack was faster, due to the inherent advantages of the generative model. However, because Par-AdvGAN introduces an autoregressive iteration mechanism in the design to improve the average success rate of adversarial attacks, this results in a certain increase in attack latency compared to AdvGAN. Therefore, we believe that it is more comparable to compare Par-AdvGAN with other optimization-based methods, which is also the main design intention of Table 7.
>
> **Advantages and innovation of Par-AdvGAN**
>
> Although the introduction of the autoregressive mechanism brings a certain speed loss, it is this mechanism that makes Par-AdvGAN significantly surpass AdvGAN in ASR. Our experimental results show that Par-AdvGAN can significantly improve the attack effect, which reflects the practical advantages of the method. In addition, the results in Table 7 further illustrate that Par-AdvGAN still achieves huge improvements in efficiency compared with other optimization-based baseline methods, demonstrating the potential of generative models and the effectiveness of our method.
>
> W2: Thank you for your suggestions of relevant literature. We highly recognize the important contribution of the methods proposed in [1][2] in enhancing the transferability of adversarial attacks. However, our research mainly focuses on improving attack performance through the optimization of generative models (such as AdvGAN). Compared with these state-of-the-art methods, our method explores a different direction - introducing an autoregressive mechanism into the generative model framework, taking into account attack efficiency and success rate. Therefore, a direct comparison of our approach with these methods may not be completely fair, as there is a significant difference in the focus of the two studies.
>
> Nonetheless, we appreciate your suggestions for these recent works and plan to further explore in future research how to combine these transferability-enhancing techniques with our framework to further improve the applicability and effectiveness of the method.
>
> W3: Thank you for your question about high processing speed application scenarios. We agree that discussing potential scenarios for real-world applications of PAR-AdvGAN better highlights the importance of speed improvements and strengthens the motivation of the article. In the following practical scenarios, the high processing speed of PAR-AdvGAN is particularly advantageous or necessary:
>
> **Large-scale data processing and automated testing**
>
> In research on adversarial attacks, testing on large-scale datasets such as ImageNet is a common task. PAR-AdvGAN is able to significantly reduce the time required to generate adversarial samples while maintaining a high attack success rate (ASR), thus speeding up the overall efficiency of model evaluation and robustness testing. This is particularly important in development environments that require rapid iteration of model designs.
>
> **Time-sensitive scenarios in security testing**
>
> In security testing scenarios that require rapid response (such as detecting system vulnerabilities or responding to urgent security threats), the speed of counterattacks may directly affect the efficiency of problem discovery and resolution. PAR-AdvGAN can generate high-quality adversarial samples in a short period of time, helping to improve the security testing efficiency of the system.
>
> **Applications on resource-constrained devices**
>
> On devices with limited computing resources (such as embedded devices or mobile terminals), rapid adversarial sample generation can reduce computing overhead and improve the actual deployment capabilities of the system. This also further expands the scope of application of PAR-AdvGAN.
>
> W4: We thank the reviewer for the suggestion and we will revise this issue.

---

### Official Review · Reviewer_npjb · 2024-11-03

**Soundness:** 2
**Presentation:** 3
**Contribution:** 3
**Rating:** 5
**Confidence:** 4

**Summary:**

This paper presents Progressive Auto-Regression AdvGAN (PAR-AdvGAN), an innovative approach that enhances the generation of adversarial examples using Generative Adversarial Networks (GANs). PAR-AdvGAN addresses this limitation by integrating an auto-regressive iteration mechanism within a progressive generation framework, resulting in adversarial examples with improved attack capabilities. The performance of PAR-AdvGAN is thoroughly evaluated through extensive experiments, showing superior results compared to various adversarial attacks and the original AdvGAN.

**Strengths:**

1. The author questions that the generation of perturbations is usually limited to a single iteration, thus degrading the attacking performance, which looks like an interesting issue.
2. The paper is well-written and easy to follow.
3. Extensive experiments are performed to evaluate the proposed PAR-AdvGAN, compared to other baselines.

**Weaknesses:**

1. The paper argues that the limitations of the previous GAN method for generating perturbations stem from its reliance on single-step sampling. While the experiments demonstrate significant improvements, I question whether this is the primary reason for the shortcomings of the previous method (probably the adversarial features are not well-learned). To further validate this assumption, what would happen if we incorporated additional time steps into the training of the GAN, such as concatenating 2 or 3 time-step images? It would be more convincing to expect further improvement.
2. This approach seems analogous to diffusion models, which are not sufficiently discussed in the paper (never mentioned even once), despite the existence of related work on adversarial attacks utilizing diffusion techniques.
3. The comparison of FPS appears somewhat unfair, as training the GAN (for 60 epochs, as mentioned in the paper) brings an additional cost that should be considered, while other gradient-based methods don't. Otherwise, the claim should only account for the time required for deployment (after training).
4. The authors raise the question RQ3 but I don't see a clear explanation why it works.
5. The proposed method PAR-AdvGAN mainly relies on the Inception and Resnet model as a source model, it would be good to explore more different architectures.

**Questions:**

1. Typo in line 330 "gaussian"
2. How to read Table 1? It seems the bold represents the result of the PAR-AdvGAN and not all of them are the best.
3. Overall the motivation to me is good and clear. However, a deeper insight would be appreciated to justify why the previous method fails and to explain the necessity of the Progressive Auto-Regression process.
4. The author mentioned AdvGAN++ but did not compare it in the experiment part. Also, are there other improved variants for AdvGAN?

---

> ### Author Response · Authors · 2024-11-24
> **Official rebuttal by authors**
>
> W1: We thank the reviewer for his suggestion. Regarding the limitations of AdvGAN, we use Figure 3 in Appendix A.1 to present the fact that AdvGAN's generator continuously increases perturbations during iterations. We believe this phenomenon is important because the goal is to maximize the attack effect with minimal perturbations. Therefore, the continuous increase in perturbations during single-step sampling of AdvGAN reflects that it has lost control of the perturbation rate, which will affect the success rate of the adversarial attack.
>
> W2: Thanks to the reviewer for the suggestion. We would like to clarify that the adversarial attack based on diffusion models is generally a slow iteration process. In addition, the diffusion model requires a complete dataset for training, while in our task scenario, the samples required to train the generator are less than 1/1000. Therefore, we believe that from the perspective of time efficiency, our method is more suitable for scenarios that require rapid generation of adversarial samples.
>
> W3: We thank the reviewer for his suggestion. We would like to clarify that 60 epochs will not increase with the increase of attack samples, that is, the additional cost of 1 hour for each model will not change with the increase of samples. While other gradient optimization-based methods will. Our experiment considers the potential performance cost. In reality, the possibility of adversarial data is hundreds or thousands of times higher than that in the experimental environment.
>
> W4: Thank you for your attention to our RQ3. We acknowledge that the explanation of the effectiveness of PAR-AdvGAN in the submission may not be clear enough, so we elaborate on the reasons behind its design and combine the ablation results to illustrate the contribution of parameter selection to the effectiveness of the method.
>
> **Theoretical basis for the effectiveness of PAR-AdvGAN**
>
> The core of PAR-AdvGAN lies in combining the efficiency of the generative model with the step-by-step optimization capability of the autoregressive iterative mechanism. This design gradually approaches the optimal adversarial sample through autoregression, avoiding the potential suboptimal solution problem of a single generation, thereby improving the attack success rate (ASR). Our method can further improve the quality of adversarial samples through iterative optimization on the basis of the fast initial solution provided by the generative model.
>
> **Support of ablation experiments**
>
> To answer RQ3 and verify the effectiveness of PAR-AdvGAN, we explored the impact of the hyperparameter λ on the performance in ablation experiments, and the results showed that the current parameter setting achieves the best balance between ASR and efficiency.
>
> W5: Thanks to the reviewer for the suggestion. In fact, we have used the ViT-B/16 model as the source model in Table 6 in row 478 to verify the performance of our method. The experimental results show that our method can still achieve the best performance overall.
>
> Q1: Thank the reviewer for pointing this out, we will revise this issue.
>
> Q2: Thanks to the reviewer for the suggestion. We would like to clarify that in most cases, our method shows a significant performance improvement in attack success rate compared with previous methods. Although DI-FGSM has a slight advantage on the target model Inc-v4 in Table 1, considering the average improvement of the seven target models, our method achieves the best results.
>
> Q3: Please refer to W1 for clarification.
>
> Q4: To the best of our knowledge, our method is the first to propose an iterative approach to generate adversarial examples, but using a generative model. Unfortunately, AdvGAN++ is not open source, so we did not compare it in our experiments.

---

### Official Review · Reviewer_d2oh · 2024-11-09

**Soundness:** 3
**Presentation:** 3
**Contribution:** 2
**Rating:** 3
**Confidence:** 4

**Summary:**

This paper extends previous GAN-based adversarial sample generation works from one-step generation to multiple-step generation. The overall technical novelty is limited, since the proposed method is a trivial extension from previous AdvGAN, simply increasing the generation step from one to $n$.

Non-trivial performance gains are shown in terms of attack success rates compared with some previous method in most cases. The exception is in Table 4, where SINI-FGSM has tiny advantage over the proposed method. Some important baseline methods (i.e. diffusion model-based attacks) are not considered in the experiments.

Overall writing is smooth and easy to follow, with some minor issues to improve for better presentation.

**Strengths:**

1. Overall writing is smooth and easy to follow.

2. The proposed method to extend AdvGAN to multiple-step attack is intuitively reasonable.

3. Non-trivial performance gains are shown in terms of attack success rates compared with some previous method in most experiments. In Table 4 SINI-FGSM has tiny advantage over the proposed method. But this is not any issue to me, since the difference is tiny enough to be ignored and also the attackers can choose the model with best generalization ability as attack-generation model and don't have to stick with the model used in Table 4.

**Weaknesses:**

1. The technical novelty is limited. The extension over AdvGAN is trivial.

2. When talking about multi-step generation, usually people would assume diffusion models to do better jobs than progressive GANs. Can diffusion models perform the same job as the proposed Par-AdvGAN? There are multiple paper applying diffusion models to generate adversarial samples like this one [1,2]. It seems to me they can be adopted for the same purposed as Par-AdvGAN. I would suggest to add them as important baseline methods in the experiments, if you find it reasonable. Otherwise, please discuss why they can't be used.

[1] Diffusion-Based Adversarial Sample Generation for Improved Stealthiness and Controllability.

[2] SD-NAE: Generating Natural Adversarial Examples with Stable Diffusion.

3. In Table 7, why not include AdvGAN when comparing attack latency? I assume AdvGAN should be faster than Par-AdvGAN. The fast attack speed actually comes from the intrinsic advantage of generative model-based attack methods over optimization-base method, instead of the novelty of Par-AdvGAN.

4. Line 9 and line 18 in table 5 is confusing. Shouldn't they be put into comments instead two extra steps of the algorithm?

**Questions:**

Please see above.

---

> ### Author Response · Authors · 2024-11-24
> **Official rebuttal by the authors**
>
> W1: We thank the reviewer for the suggestion. To the best of our knowledge, our approach is the first to suggest the iterative generation of adversarial examples, but using a generative model.
>
> W2: Thanks to the reviewer for the suggestion. We would like to clarify that the adversarial attack based on diffusion models is generally a slow iteration process. In addition, the diffusion model requires a complete dataset for training, while in our task scenario, the samples required to train the generator are less than 1/1000. Therefore, we believe that from the perspective of time efficiency, our method is more suitable for scenarios that require rapid generation of adversarial samples.
>
> W3: Regarding the non-included comparison of AdvGAN and its attack speed in Table 7, we provide the following explanation:
>
> **Why AdvGAN is not included in Table 7**
>
> We did notice that the AdvGAN attack was faster, due to the inherent advantages of the generative model. However, because Par-AdvGAN introduces an autoregressive iteration mechanism in the design to improve the average success rate of adversarial attacks, this results in a certain increase in attack latency compared to AdvGAN. Therefore, we believe that it is more comparable to compare Par-AdvGAN with other optimization-based methods, which is also the main design intention of Table 7.
>
> **Advantages and innovation of Par-AdvGAN**
>
> Although the introduction of the autoregressive mechanism brings a certain speed loss, it is this mechanism that makes Par-AdvGAN significantly surpass AdvGAN in ASR. Our experimental results show that Par-AdvGAN can significantly improve the attack effect, which reflects the practical advantages of the method. In addition, the results in Table 7 further illustrate that Par-AdvGAN still achieves huge improvements in efficiency compared with other optimization-based baseline methods, demonstrating the potential of generative models and the effectiveness of our method.
>
> W4: We are sorry that we do not quite understand what the reviewer meant by rows 9 and 18. We did not find it in Table 5. We would like to ask if the reviewer made a mistake in citing it?

---

> > ### Comment · Reviewer_d2oh · 2024-11-30
> >
> > Thank you for your effort in providing a response. However, I did not find additional information beyond what I was already aware of during my initial review. Specifically, my concerns regarding the limited novelty and the absence of experiments comparing with diffusion-based attacks remain unaddressed. As a result, my score will remain unchanged.
> >
> > Here are some more feedbacks on your response.
> >
> > 1. Diffusion models can also be fast during inference. For example, [1] shows it can achieve image generation on LSUN dataset using 10 steps sampling during inference. There are many more related works on this thread.
> >
> > [1] Towards More Accurate Diffusion Model Acceleration with A Timestep Tuner. CVPR 2024.
> >
> > 2. Sorry for the confusion. I mean line 9 and 18 in Algorithm 1 should be put into comments. This is a minor issue regarding presentation. Not much affecting with my evaluation score to the paper.

---

> > > ### Author Response · Authors · 2024-12-03
> > > **Further clarification**
> > >
> > > We thank the reviewer for the comment. However, we regret to note that the comment lacks constructive suggestions, particularly regarding the Diffusion-based model.
> > >
> > > Regarding [1] and [2], we acknowledge and appreciate their research efforts in crafting novel attack methods. As noted in their context, their approach emphasizes **"combining the strong prior knowledge of the diffusion model into adversarial sample generation."** However, these works **do not include comparisons with AdvGAN**, focusing instead on other baseline methods such as PGD, AdvPatch, and AdvCam.
> > >
> > > From a methodological perspective, our approach is fundamentally different, introducing a novel iterative generation process for adversarial examples. We also assess the efficiency of our method with the proposed auto-regressive iteration mechanism. Nonetheless, we are happy to consider incorporating a discussion of diffusion-based models in our future work. **Given the extensive volume of adversarial attack research, we believe our experimental design is sufficiently broad to clarify our contributions and validate our performance improvements.**
> > >
> > > Furthermore, we agree with the reviewer’s acknowledgment that **"Non-trivial performance gains are shown."** This reinforces our belief that our contributions are meaningful and significant. We sincerely hope that reviewers evaluate each work on its intrinsic merits and recognize the value of our contributions.

---

### Meta-Review · Area_Chair_4VHi · 2024-12-22

**Metareview:**

This paper presents an enhancement of AdvGAN, which incorporates an auto-regressive iteration mechanism for enabling the multi-step generation of adversarial examples. The provided experiments show that AdvGAN shows significant improvements in attack success rates and generation speed compared to existing methods.

The reviewers found the technical design to be well-motivated, and the overall presentation was clear and easy to follow. However, several major concerns were raised: 1) the technical contribution of the paper is somewhat limited; 2) missing comparisons with more advanced adversarial attacks, particularly diffusion model-based attacks and recent transfer attack techniques; and 3) additional ablation studies and analyses are needed, like experiments on other datasets (e.g., CIFAR-10/100) and careful examination of different design components.

The rebuttal and follow-up discussions were considered, but due to the absence of additional experiments, the reviewers remain unconvinced. As a result, all reviewers have voted against accepting the paper. The AC supports this decision, and strongly encourages the authors to address these concerns more thoroughly in order to prepare a stronger submission in the future.

**Additional Comments On Reviewer Discussion:**

The major concerns are summarized in my meta-review. Given the limited empirical results provided in the rebuttal, the reviewers remain unconvinced regarding points (2) and (3). The AC agrees that these concerns are legitimate and must be carefully addressed to fully demonstrate the effectiveness and the advantage of the developed PAR-AdvGAN. Therefore, the final decision is rejection.

---

### Decision · Program_Chairs · 2025-01-22

Reject